# Anemone Cut Flower Timing, Yield, and Quality in a High-Elevation Field and High Tunnel

Shannon Rauter [1,*], Melanie Stock [1], Brent Black [1], Dan Drost [1], Xin Dai [2] and Ruby Ward [3]

1 Department of Plants, Soils and Climate, Utah State University, 4820 Old Main Hill, Logan, UT 84322, USA
2 Utah Agricultural Experiment Station, Utah State University, 4810 Old Main Hill, Logan, UT 84322, USA
3 Department of Applied Economics, Utah State University, 4835 Old Main Hill, Logan, UT 84322, USA
* Correspondence: shannon.rauter@usu.edu

**Abstract:** A narrow window of optimal spring temperatures limits anemone (*Anemone coronaria* L.) cut flower production in the US Intermountain West, where fall plantings risk winter injury and spring plantings are limited by summer dormancy. Regional management recommendations are needed to improve anemone harvest timing and yield for growers in USDA hardiness zones 6 and below (average annual minimum temperatures below −18 °C). The aim of this research was to optimize flower timing, yield, quality, and profitability in high tunnel and field production systems by evaluating planting dates, winter insulation, tuber preparation, and cultivar selection. High tunnel and field trials were conducted from fall 2020 to spring 2022 in North Logan, UT (41.767° N, −111.811° W, 1405 m elevation, USDA hardiness zone 5). Tubers were pre-sprouted or directly planted into a high tunnel (left bare or covered with low tunnels) or field (left bare or covered with mulch, a low tunnel, or mulch and a low tunnel) from November to April. Harvest began as early as 2 March in the high tunnel and 9 April in the field, with overall average marketable yields (stems per m$^2$ ± SE) of 142 ± 7 in the high tunnel and 85 ± 4 in the field. Planting pre-sprouted tubers under low tunnels in the high tunnel in November delivered the earliest harvest (2 March), greatest marketable yield (280 stems per m$^2$ ± 73 SE), and greatest net returns ($38 per m$^2$). For November field plantings, insulation improved emergence by 75% and marketable yield by 77 stems per m$^2$ ± 15 SE. Combining high tunnel and field production with the season advancement techniques of fall planting dates, low-cost insulation, and pre-sprouting resulted in high total yields in the Intermountain West compared to traditional industry recommendations.

**Keywords:** *Anemone coronaria*; 'Carmel'; 'Galilee'; high tunnel; quality; yield





## 1. Introduction

The growing demand for local agricultural products, including cut flowers, has spurred an increase in the number of specialty cut flower growers nationwide over the past decade, as well as within the US Intermountain West Region (Utah, Idaho, Nevada, and western Colorado) [1–3]. Of 188 North American cut flower growers surveyed in 2017, 95% were small growers (<10 employees) and only 15% were farming in the Western US (West, Northwest, Southwest, and Northern Rockies and Plains climate regions [4]), highlighting that most management recommendations for small cut flower farms are targeted to milder climates that constitute the majority of production [5]. Anemone (*Anemone coronaria* L.) is an economically important crop that was produced by 40.0% of surveyed North American cut flower growers in 2017 [1,5]. The total yield potential for anemone is over 20 stems per plant based on research in the Eastern Mediterranean where the anemone originated [6,7]. In the US, total yields from field and high tunnel studies typically range from 3 to 7 stems per plant [6,8–10], with the greatest average total yield reported at 10 stems per plant in a fall-planted Georgia field trial (USDA hardiness zone 8) [10]. The stem length potential is up to 60 cm [6,11], but US field and high tunnel trials reported stem lengths ranging

from 10 to 30 cm [8–10] and Israeli high tunnel trials reported that approximately 20% of stems produced were too short to market (<20 cm) [6]. In North America, stem length was listed as the most common production challenge among surveyed growers in 2017 [5]. The price for one bunch of ten anemone stems ranged from $10.00 to $20.00 at the Boston terminal market and $12.00 to $17.00 in Northern Utah in 2021, with prices varying by country of import, month, and stem length [12,13]. High potential total yield and profit make anemone a promising specialty cut flower crop, creating a need for regionally specific recommendations to increase stem lengths and marketable yields for small growers in the Intermountain West [14].

Anemones concentrate growth when the weather is cool and wet, with optimal temperatures for growth and flowering ranging from 5 to 10 °C at night and 12 to 18 °C during the day [15,16]. Flowering typically occurs after five to six leaves have been produced, approximately three months after planting, and continues until dormancy is initiated by a combination of high temperatures (>25 °C) and long days (>12 h), with a larger effect from temperature than photoperiod [7,17]. Anemones are typically forced from tubers that are replaced annually for small farm production, and are often grown with ranunculus (*Ranunculus asiaticus L.*) due to their similar temperature requirements [10,18,19]. Grower resources advise fall planting in USDA hardiness zones 7 and above, and spring planting for USDA hardiness zones 6 and below [15,18] based on the minimum survival threshold of −3 to −4 °C for hydrated tubers (80–90% water content) [20–22]. However, recommendations based on air temperature may be poor hardiness indicators for tubers that are responding to soil temperature, which is also influenced by snow patterns, before emergence [23,24]. Additionally, some grower resources indicate that anemone tubers may be more cold-hardy than ranunculus tuberous roots, tolerating temperatures as low as −7 °C, but this information is largely based on informal observations [18,19].

The second most common anemone production challenge in a survey of North American growers was timing, including a short harvest window [5]. In the Intermountain West, optimal anemone temperature ranges occur over a window of roughly three months in the spring [17,25–27]. Though fall planting is not recommended because of the risk of cold injury, spring plantings are limited by early summer heat. Dormancy can be expected to begin by late June in Northern Utah, when daily maximum temperatures begin reaching 25 °C and daylengths are over 15 h [27]. Concentrating anemone production when air temperatures range from 5 to 18 °C, which typically occurs from March to May, will help Intermountain West growers maximize production before plants go dormant [28]. Advancing harvest as early as possible also offers additional benefits since insect and weed pressure increase, demand for cool-season cut flowers decreases [15,29], and water availability can limit production [30] as temperatures increase over the growing season.

Fall planting coupled with the use of insulation, such as high tunnels, low tunnels, and mulch, has the potential to advance growth and flowering, while protecting tubers from cold injury. High tunnels are passively heated and cooled structures that can increase air temperature from between 10 °C (with venting) to 30 °C (without venting) compared to the outside air on a sunny day [31,32]. The protected environment of a high tunnel also allows for winter planting, roughly one month of growing season advancement, and improved yields and stem lengths compared to field production [33–38]. Despite high tunnel benefits, many Utah cut flower farms are considered micro-farms and are limited to field production due to lack of space [39]. Low tunnels and mulches, such as straw, present an alternative to insulate fall-planted tubers from subfreezing temperatures and reduce costs compared to high tunnel production [15,23,34,40]. Low tunnels covered with fabric row cover or greenhouse plastic can increase average daily maximum temperatures by up to 3 °C (soil) to 18 °C (air) [41,42]. Mulch can provide further insulation of the soil. With 15 cm of straw mulch, soil temperature at a 15 cm depth was 4.2 °C warmer, on average, than soil with only natural snow cover in a Minnesota trial [40].

Pre-sprouting tubers rather than direct planting and cultivar selection can also advance growth and optimize flower timing, quality, and marketable yield. Pre-sprouting

consists of starting hydrated tubers in a growing medium at a cool temperature for two to five weeks [15,43,44]. Four weeks at 5 °C hastened anemone flowering by approximately 20 days and improved average total yield from 7 to 8 stems per plant in a Japanese greenhouse study [16], while three weeks at 10 °C improved winter survival of anemone in a New York high tunnel and did not significantly impact yield [45]. The timing of flowering, as well as stem length, bloom diameter, and yield can also be impacted by cultivar [8,10], indicating the need to select cultivars optimized for regional production. Since most previous anemone research was conducted in greenhouses [16,17] or warmer climates [6,7,9,10], additional research is needed to develop recommendations that deliver consistent results for both field and high tunnel growers in USDA hardiness zones 6 and below (average annual minimum temperatures below −18 °C). The objective of this study was to evaluate anemone emergence, timing, yield, and quality by (1) fall versus spring planting dates; (2) combinations of the winter insulation methods of high tunnels, low tunnels, and straw mulch; (3) pre-sprouting tubers before planting versus direct planting; and (4) cultivar. We hypothesized that pre-sprouting, fall planting, and insulating tubers increase anemone yield and quality by advancing growth and production during more optimal, early-season temperatures compared to non-pre-sprouted, uninsulated spring plantings.

## 2. Materials and Methods

Trials were conducted from November to July in 2020-21 and 2021-22 in a field and high tunnel located approximately 50 m apart at the Utah Agricultural Experiment Station Greenville Research Farm in North Logan, UT (lat. 41.77° N, long. 111.81° W, 1382 m elevation, 135 freeze-free days, an average last frost date on 15 May, and USDA hardiness zone 5). The soil is a Millville silt loam with 2% organic matter [46]. In the field, three 1.2 m W $\times$ 12.2 m L beds were each subdivided into four, 3.7 m$^2$ plots (whole plots). Whole plots were randomly assigned an insulation treatment: low tunnel with mulch (+LT + M), low tunnel only (+LT − M), mulch only (−LT + M), and bare soil (−LT − M); and then divided into three subsections and randomly assigned a planting date (15 to 18 November, 17 to 22 March, and 16 to 18 April). Rows within each subsection were assigned a cultivar ('Carmel' and 'Galilee') and a pre-sprouting treatment (pre-sprouted [+PS] and non-pre-sprouted [-PS]). One 4.3 m W $\times$ 12.8 m L high tunnel [47] oriented east-west was subdivided into six 4.4 m$^2$ plots (whole plots), organized in a randomized complete block design. The addition (+LT) and absence of a low tunnel (−LT) were tested in triplicate at the whole plot level with a subplot treatment factor of pre-sprouting (+PS and −PS). The treatment factors of planting date (16–18 November, 18–20 January, 14–17 February, and 16–17 March) and cultivar ('Carmel' and 'Galilee') were randomly assigned to rows within subplots.

Before planting in November each year, the high tunnel and field were rototilled to an approximate depth of 0.10 to 0.15 m. Nitrogen (N) fertilizer (guaranteed analysis 46N-0P-0K) was incorporated as a split application at a rate of 73 kg N ha$^{-1}$ in November and again in spring when flower buds were visible, while Phosphorus (P, 0-20-0) and Potassium (K, 0-0-50) fertilizer applications were based on soil test results. Color mixes of anemone 'Carmel' and 'Galilee' (size 5–6 cm) tubers (Ball Horticulture, Chicago, IL, USA) were soaked in 15 to 25 °C running tap water for 4 h [43]. The tubers were drained and submerged in a 0.3% Captan fungicide solution for the last hour of soaking to reduce risk of rot [43]. Pre-sprouted (+PS) tubers were placed in flats with a moist soilless growing medium consisting of 68% peat moss (Canadian sphagnum peat moss; Sun Gro Horticulture, Agawam, MA, USA), 32% vermiculite (Therm-O-Rock West, Chandler, AZ, USA), 0.48 g·L$^{-1}$ AquaGro 2000 G (Aquatrols, Paulsboro, NJ, USA), and 1.43 g·L$^{-1}$ hydrated lime (Ca(OH)$_2$; Mississippi Lime, St. Louis, MO, USA). The pre-sprouted tubers were kept at a day/night temperature regime of 22/18 °C for one week, followed by an average temperature of 11.7 °C for an additional week to acclimate to cooler temperatures. Tubers were planted 0.15 m apart within and between rows at a depth of 0.05 m, and drip tape

(Aqua-Traxx, Toro, Bloomington, MN, USA) was used to irrigate zero to three times per week, depending on environmental demand.

High tunnel temperatures were managed throughout the season by manually venting the structure based on field weather conditions reported from an automatic weather station located 0.2 km away [28] according to [36]. Low tunnels in the high tunnel and field were covered with fabric row cover (AG-50, 50.6 $g \cdot m^{-2}$, Arbico Organics, Oro Valley, AZ, USA), and manually vented when ambient air temperature was over 15 °C [39]. Mulch consisted of approximately 0.1 m of straw placed on the soil surface (approximately 2 $kg \cdot m^{-2}$). Based on air temperature monitoring each year, mulch was removed between 1 and 15 March, low tunnels were removed between 13 April and 11 May, field plots were shaded with 30% shade fabric (DeWitt, Woven Shadecloth Fabric, Sikeston, MO, USA) between 23 April and 11 May, and high tunnel plastic was replaced with 30% shade fabric between 5 and 13 May.

A shielded thermistor (CS 107, Campbell Scientific, Logan, UT, USA) measured air temperature at 0.25 m in one replicate per field and high tunnel insulation treatment. Additional air temperature data (2.0 m height) was collected from an automatic weather station located 0.2 km away [28]. One soil temperature and moisture sensor (True-TDR-315H, Acclima, Inc., Meridian, ID, USA) was installed at a 0.05 m depth in one replicate per field insulation treatment and three replicates per high tunnel insulation treatment. Data loggers (CR-1000, Campbell Scientific) and multiplexers (AM25T, Campbell Scientific) recorded data at one-minute intervals to calculate hourly and daily averages. Precipitation was measured at the Utah State University campus approximately 3 km from the trial [48].

Plant emergence was counted weekly from planting to harvest, with a plant counted when a shoot was visible above the soil surface. Harvest occurred three to four times per week, with stems cut when blooms were fully colored and had opened and closed once [43]. Local farms reported market preferences and pricing that were used to grade stems by length and quality. Quality grade stems were at least 25 cm long, free of curvature or visual deformities, and sold for $15 per bunch of ten stems ($1.50 per stem); speculation grade stems were 20–25 cm long, free of curvature or visual deformities, and sold for $12.50 per bunch of ten stems ($1.25 per stem); and cull grade stems were shorter than 20 cm, curved or deformed, and not marketable. An economic budget was calculated based on cost, yield, and sales by management practice (planting $\times$ insulation $\times$ pre-sprouting averaged across both cultivars) for a 52 $m^2$ high tunnel or field production area containing 994 plants. All input costs for production were recorded [49] and stems were sold wholesale from April to July through a local cut flower co-op that marketed to florists in Logan, Salt Lake City, and Park City, Utah. Crop value was calculated by scaling the average yield of quality grade and speculation grade stems for each combination of management practices to a whole high tunnel or field and multiplying by the price and the percent expected to sell to florists (assumed to be 100% for quality grade and 50% for speculation grade stems). Net returns were calculated as the difference between crop value and input costs.

Yield (stems per $m^2$) was calculated by dividing the number of stems harvested by the number of tubers planted in one replication and multiplying by a plant density of 44 plants per $m^2$, without accounting for plant emergence. First harvest was considered the date of harvest of the first flower, regardless of its quality grade (i.e., total yield). The times to reach 20% (T20), 50% (T50), and 80% (T80) of marketable yield were calculated, with T50 representing the harvest midpoint and the time from T20 to T80 (T20-80) representing the duration of peak harvest. An ANOVA-type mixed model was used to compare emergence, production timing, total yield, and marketable yield among insulation, planting date, pre-sprouting, and cultivar treatments within the high tunnel and field. Percentages of emergence were logit transformed, while total yield, marketable yield, and T20-80 were log transformed. The proportions of each of the three stem quality grades were compared across treatments with a mixed model on categorical outcomes. All statistical analyses were performed with PROC GLIMMIX of SAS Studio (SAS Institute, Cary, NC, USA) at a significance level of $\alpha$ = 0.05.

Grower collaborators collected additional winter survival and yield data along Utah's Wasatch Front from 2020-22. Eleven growers participated in 2020-21 and seven growers participated in 2021-22, from across Cache, Weber, Davis, Salt Lake, and Utah counties (USDA hardiness zones 5 to 7). Participants ranged from skilled hobbyists to micro-farmers with no to moderate experience growing anemone and were provided ten anemone 'Carmel' tubers to plant each fall and spring. Growers were instructed to plant at a 0.05 m depth and record management decisions, including planting dates, winter insulation, pre-sprouting, fertilization, and irrigation.

## 3. Results

### 3.1. Environmental Conditions (15 November–15 July)

Total precipitation (rainfall and a snow water equivalent) from 15 November to 15 July was 153 mm in 2020-21 and 178 mm in 2021-22, which were less than the 30-year normal (1981–2010) of 359 mm [27]. Total snowfall was 762 mm in 2020-21 and 516 mm in 2021-22 (Figure 1), and total solar radiation was 4195 MJ·m$^{-2}$ in 2020-21 and 4044 MJ·m$^{-2}$ in 2021-22. In both years, monthly average air temperatures were near the 30-year normals across the study period [27]. From 15 November to 1 March, the average air temperature (2 m height) was −2.2 °C in 2020-21 and −2.9 °C in 2021-22, and from 1 March to 15 July, the average air temperature was 13.3 °C in 2020-21 and 11.5 °C in 2021-22. The daily average air temperature first reached 25 °C on 5 June 2021 and 17 June 2022. In the high tunnel, air temperature (0.25 m height) was 2.3 ± 0.0 °C greater in +LT than −LT in 2020-21 and 2.8 ± 0.0 °C greater in 2021-22 (Figure 2).

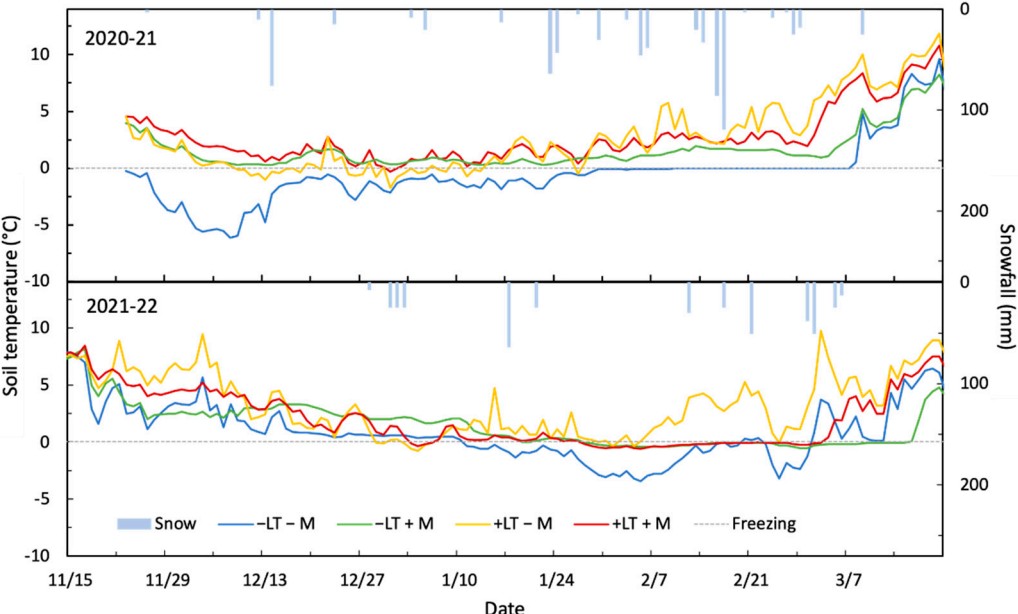

**Figure 1.** Average daily snowfall (light blue bars) and soil temperature (0.05 m depth) during the winter of 2020-21 (top) and 2021-22 (bottom) under no cover (−LT − M; blue line), mulch (−LT + M; green line), a fabric low tunnel (+LT − M; yellow line), and mulch and a fabric low tunnel (+LT + M; red line) in the field in North Logan, UT.

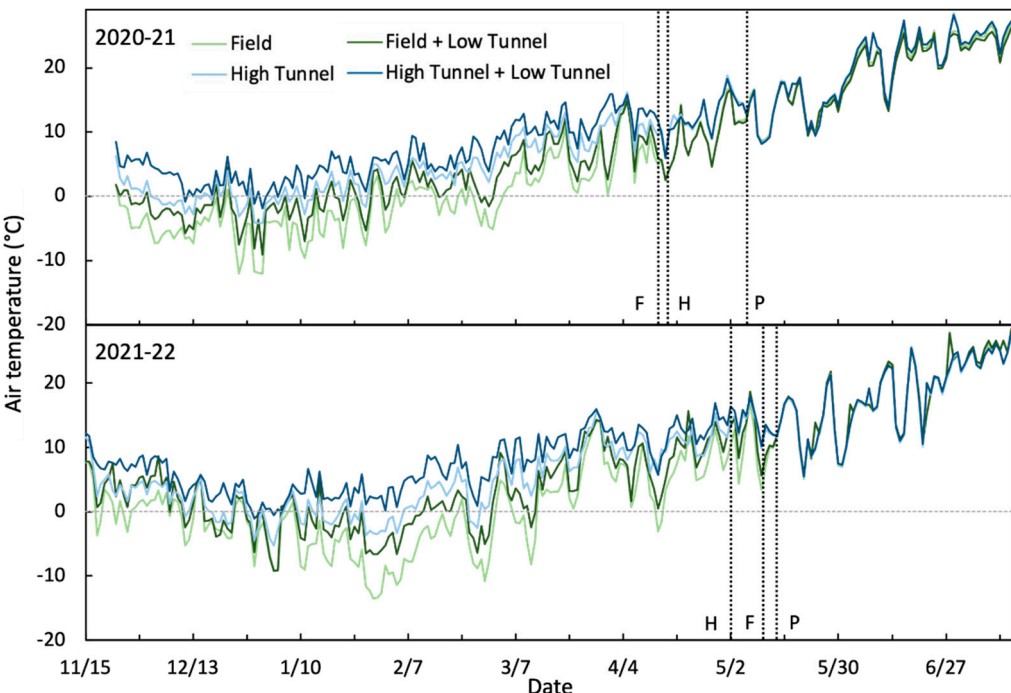

**Figure 2.** Average daily air temperature (0.25 m) in 2020-21 (top) and 2021-22 (bottom) in an uninsu-lated field (light green line), a field covered with low tunnels (dark green line), a high tunnel (light blue line), and a low tunnel within a high tunnel (dark blue line) in North Logan, UT. Vertical dashed lines marked with F, H, and P denote the dates on which field low tunnels, high tunnel low tunnels, and high tunnel plastic were removed, respectively.

Bare soil ($-$LT $-$ M) was below 0 °C from 23 November 2020 to 6 March 2021 and 11 January 2021 to 2 March 2022 (Figure 1). The minimum bare soil temperature was $-9.2$ °C (9 December) in 2021 and $-6.9$ °C (24 February) in 2022, while the total time bare soil temperature was at or below $-3.0$ °C was 325 h in 2020-21 and 182 h in 2021-22. Compared to bare soil each year, the average hourly soil temperature was 0.8 to 2.4 °C greater for $-$LT $+$ M, 2.2 to 2.9 °C greater for $+$LT $-$ M, and 0.9 to 2.8 °C greater for $+$LT $+$ M. In the high tunnel, soil temperature was 0.8 $\pm$ 0.0 °C greater in $+$LT than $-$LT in 2020-21 and 1.0 $\pm$ 0.0 °C greater in 2021-22. Soil temperature was below 0 °C for a total of 0 h for $+$LT and 24 h for $-$LT in 2020-21 and 18 h for $+$LT and 114 h for -LT in 2021-22.

*3.2. Shoot Emergence*

In the field, emergence was significantly affected by the main effects of *cultivar* ($p < 0.0001$), *pre-sprouting* ($p = 0.0260$), *insulation* ($p = 0.0220$), and *planting date* ($p = 0.0169$), as well as by the interaction effect of *insulation × planting date* ($p < 0.0001$). Emergence averaged 93 $\pm$ 1% for 'Galilee' compared to 87 $\pm$ 1% for 'Carmel' and 92 $\pm$ 1% for $+$PS com-pared to 88 $\pm$ 2% for -PS. Emergence for the November planting was 55 $\pm$ 6% for $-$LT $-$ M tubers compared to 96 $\pm$ 2% for tubers with any form of insulation ($+$LT $-$ M, $-$LT $+$ M, or $+$LT $+$ M; $p < 0.0001$), while for March and April plantings, emergence was 92 $\pm$ 7% across all insulation treatments. Emergence of $-$LT $-$ M tubers planted in November was 69 $\pm$ 6% in 2020-21 and 41 $\pm$ 9% in 2021-22. High tunnel emergence across both years and all management practices tested was 94 $\pm$ 1% and was only significantly affected by the main effect of *cultivar* ($p = 0.0015$), with 96 $\pm$ 1% emergence for 'Galilee' compared to 93 $\pm$ 1% for 'Carmel.'

*3.3. Harvest Timing*

In the field, first harvest and the harvest midpoint (T50) were significantly affected by the main effects of *year*, *pre-sprouting*, *planting date*, and *insulation*, as well as by the

interaction effect of *insulation × planting date*. Across all planting dates, first harvest occurred 5 ± 1 days sooner in 2021 than in 2022 (*p* = 0.0047) and 5 ± 1 days sooner for +PS than for −PS (*p* < 0.0001). First harvest occurred on 9 and 21 April for November plantings, 8 and 10 May for March plantings, and 23 and 18 May for April plantings in 2021 and 2022, respectively (Figure 3). First harvest of November plantings occurred 21 ± 2 days sooner with insulation compared to −LT − M (*p* < 0.0001). T50 occurred 3 days sooner in 2021 than in 2022 (*p* = 0.0474), occurring on 28 May for November plantings, 8 June for March plantings, and 22 June for April plantings on average across both years. The duration of peak harvest (T20-80) averaged 18 ± 0 days and was 5 ± 1 days longer in 2022 than in 2021 (*p* = 0.0027) and 10 ± 1 days longer for November plantings with insulation compared to -LT-M (*p* < 0.0001). Harvest ended between 12 and 13 July for all planting dates.

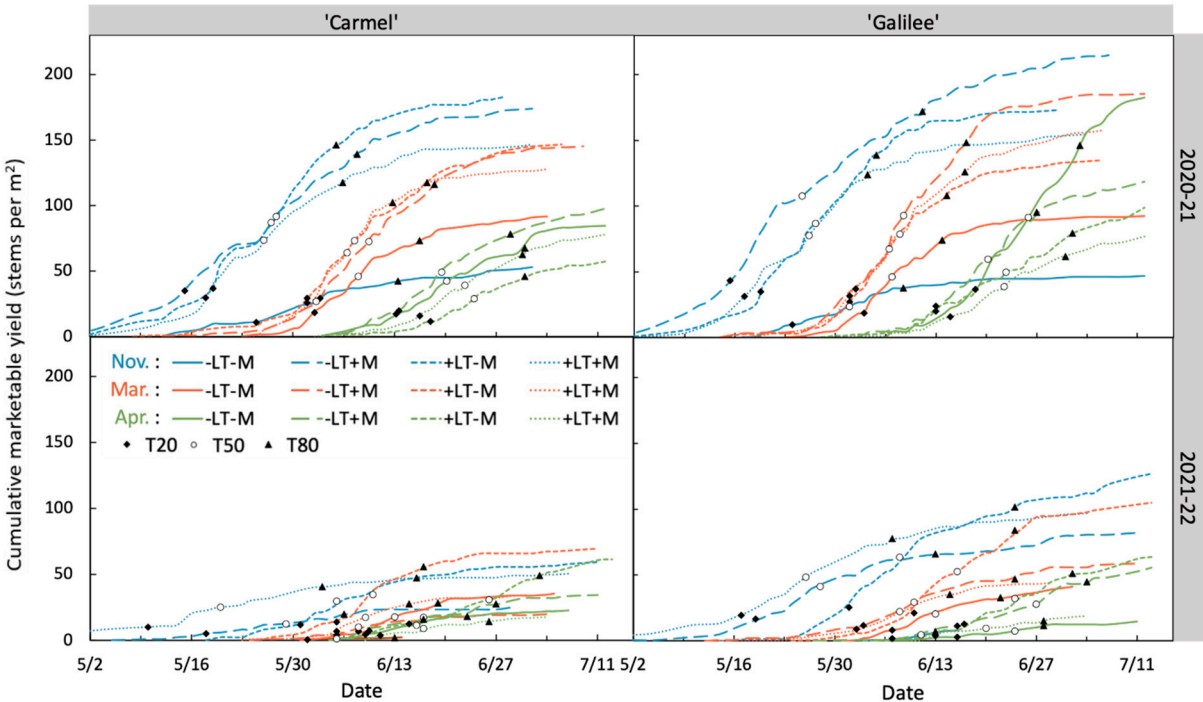

**Figure 3.** Field cumulative marketable yield of 'Carmel' (**left**) and 'Galilee' (**right**) anemone in 2020-21 (**top**) and 2021-22 (**bottom**) in North Logan, UT, by planting date (November—blue; March—red; April—green) that were left uncovered (−LT − M—solid) or insulated with mulch (−LT + M—long dashes), a low tunnel (+LT − M—short dashes), or mulch and a low tunnel (+LT + M—dots). T20 (diamonds), T50 (circles), and T80 (triangles) mark when 20, 50, and 80% of the total marketable yield is achieved, respectively.

In the high tunnel, first harvest was significantly affected by the main effects of *planting date* (*p* < 0.0001) and *pre-sprouting* (*p* = 0.0040), as well as by the interaction effects of *pre-sprouting × planting date* (*p* < 0.0001) and *low tunnels × planting date* (*p* = 0.0019). First harvest occurred on 9 and 2 March for November plantings, 23 March and 1 April for January plantings, 4 and 11 April for February plantings, and 22 and 24 April for March plantings in 2021 and 2022, respectively (Figure 4). First harvest was 8 ± 1 days earlier for +PS than for −PS for February and March plantings (*p* < 0.0001). T50 was significantly affected by the main effects of *year* (*p* = 0.0165), *low tunnels* (*p* = 0.0382), *planting date* (*p* < 0.0001), and *pre-sprouting* (*p* = 0.0209). T50 occurred 6 ± 1 days sooner in 2021 than in 2022 and 4 ± 2 days sooner for +LT than for −LT. T50 occurred on 29 April for November plantings, 7 May for January plantings, 21 May for February plantings, and 6 June for March plantings on average, across both years. T20-80 was 37 ± 2 days for November plantings, 25 ± 1 days for January plantings, 26 ± 1 days for February plantings, and 22 ± 1 days for March plantings. T20-80 was 11 ± 1 days longer in 2022 than in 2021 (*p* = 0.0032), 4 ± 1 days longer for +LT

than for −LT ($p = 0.0220$), and 4 ± 1 days longer for 'Galilee' than for 'Carmel' ($p = 0.0074$). Last harvest occurred on 30 June 2021 and 11 July 2022.

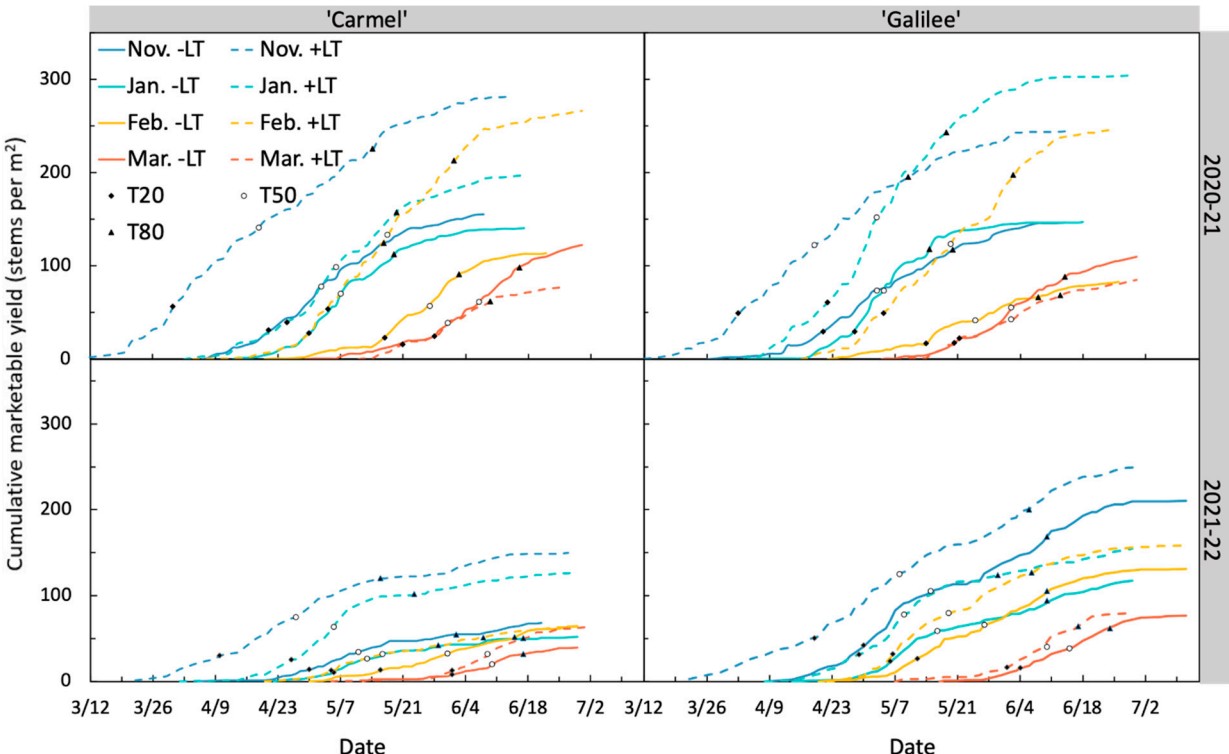

**Figure 4.** High tunnel cumulative marketable yield of 'Carmel' (**left**) and 'Galilee' (**right**) anemone in 2020-21 (**top**) and 2021-22 (**bottom**) in North Logan, UT, by planting date (November—blue; January—aqua; February—yellow; March—red) that were covered with fabric low tunnels (+LT; dashed) or left bare (−LT; solid). T20 (diamonds), T50 (circles), and T80 (triangles) mark when 20, 50, and 80% of the total marketable yield is achieved, respectively.

### 3.4. Field Yield

Total and marketable yield were affected by all main effects (*year, planting date, insulation, cultivar*, and *pre-sprouting*), as well as by the interaction effects of *insulation* × *planting date* and *pre-sprouting* × *planting date* × *cultivar* ($p < 0.05$). The simple effects of *planting date, insulation*, and *pre-sprouting* at each level of the other factors for total and marketable yield are presented in Table 1. Average total field yields ranged from 31 to 259 stems per m$^2$, with an overall average yield of 151 ± 5 stems per m$^2$, while average marketable yields ranged from 8 to 147 stems per m$^2$, with an overall average of 85 ± 4 stems per m$^2$ (Figure 3). Marketable yield was significantly greater in 2021 (102 ± 12 stems per m$^2$) than in 2022 (32 ± 4 stems per m$^2$; $p = 0.0022$). For tubers planted in November, marketable yield was 77 ± 15 stems per m$^2$ greater with any type of insulation compared to bare soil ($p < 0.0001$). No significant yield differences by insulation were observed for March or April plantings, although any type of insulation tended to increase marketable yield for March plantings. Marketable yield was 64 ± 6 stems per m$^2$ for the November planting, 68 ± 7 stems per m$^2$ for the March planting, and 44 ± 5 stems per m$^2$ for the April planting. Marketable yield was 17 ± 5 stems per m$^2$ greater for 'Galilee than for 'Carmel' ($p = 0.0006$) and 25 ± 5 stems per m$^2$ greater for +PS than for −PS ($p < 0.0001$).

**Table 1.** Field anemone production by calculated mean (±SE) total and marketable yields in stems per m$^2$ by cultivar ('Carmel' and 'Galilee'), pre-sprouting (pre-sprouted +PS and non-pre-sprouted −PS), insulation (no insulation −LT − M, mulch −LT + M, low tunnel +LT − M, and low tunnel and mulch +LT + M), and planting date (November, March, and April) in North Logan, UT, over 2020-21 and 2021-22.

| Cultivar | PS | Insulation | Total Yield (Stems per m$^2$) | | | Marketable Yield (Stems per m$^2$) | | |
|---|---|---|---|---|---|---|---|---|
| | | | **Planting Date** | | | | | |
| | | | November | March | April | November | March | April |
| 'Carmel' | +PS | −LT − M | 51 ± 10 [B,b,X] | 123 ± 21 [A,a,X] | 138 ± 24 [A,a,X] | 20 ± 7 [B,b,X] | 55 ± 15 [A,a,X] | 57 ± 15 [A,a,X] |
| | | −LT + M | 197 ± 34 [A,a,X] | 218 ± 38 [A,a,X] | 142 ± 25 [A,a,X] | 84 ± 23 [AB,a,X] | 139 ± 38 [A,a,X] | 57 ± 15 [B,a,X] |
| | | +LT − M | 140 ± 24 [A,a,X] | 158 ± 27 [A,a,X] | 173 ± 30 [A,a,X] | 64 ± 17 [A,a,X] | 71 ± 19 [A,a,X] | 72 ± 19 [A,a,X] |
| | | +LT + M | 172 ± 30 [A,a,X] | 126 ± 22 [A,a,X] | 120 ± 21 [A,a,X] | 78 ± 22 [A,a,X] | 65 ± 18 [A,a,X] | 45 ± 12 [A,a,X] |
| | −PS | −LT − M | 43 ± 8 [B,b,X] | 100 ± 17 [A,a,X] | 73 ± 13 [AB,a,Y] | 20 ± 7 [A,b,X] | 49 ± 13 [A,a,X] | 23 ± 7 [A,a,Y] |
| | | −LT + M | 198 ± 34 [A,a,X] | 116 ± 20 [AB,a,Y] | 77 ± 13 [B,a,Y] | 84 ± 23 [A,a,X] | 44 ± 12 [AB,a,Y] | 26 ± 7 [B,a,Y] |
| | | +LT − M | 148 ± 25 [A,a,X] | 89 ± 15 [AB,a,Y] | 81 ± 14 [B,a,Y] | 61 ± 16 [A,a,X] | 42 ± 14 [A,a,X] | 27 ± 7 [A,a,Y] |
| | | +LT + M | 153 ± 27 [A,a,X] | 113 ± 20 [A,a,X] | 53 ± 9 [B,a,Y] | 73 ± 20 [A,a,X] | 49 ± 14 [A,a,X] | 29 ± 10 [A,a,X] |
| 'Galilee' | +PS | −LT − M | 65 ± 12 [B,b,X] | 178 ± 31 [A,a,X] | 141 ± 30 [A,a,X] | 37 ± 12 [A,b,X] | 78 ± 21 [A,a,X] | 44 ± 14 [A,ab,X] |
| | | −LT + M | 259 ± 45 [A,a,X] | 236 ± 41 [A,a,X] | 177 ± 33 [A,a,X] | 147 ± 40 [A,a,X] | 144 ± 39 [A,a,X] | 98 ± 29 [A,a,X] |
| | | +LT − M | 212 ± 37 [A,a,X] | 215 ± 37 [A,a,X] | 191 ± 40 [A,a,X] | 140 ± 38 [A,a,X] | 129 ± 35 [A,a,X] | 94 ± 31 [A,a,X] |
| | | +LT + M | 189 ± 33 [A,a,X] | 185 ± 32 [A,a,X] | 117 ± 22 [A,a,X] | 102 ± 28 [A,ab,X] | 99 ± 27 [A,a,X] | 23 ± 7 [B,b,Y] |
| | −PS | −LT − M | 31 ± 6 [B,b,Y] | 88 ± 15 [A,a,Y] | 101 ± 21 [A,a,X] | 8 ± 3 [B,b,Y] | 32 ± 9 [A,a,Y] | 19 ± 6 [AB,b,Y] |
| | | −LT + M | 211 ± 36 [A,a,X] | 159 ± 27 [AB,a,X] | 99 ± 19 [B,a,Y] | 121 ± 33 [A,a,X] | 79 ± 21 [A,a,X] | 52 ± 15 [A,ab,X] |
| | | +LT − M | 192 ± 33 [A,a,X] | 135 ± 23 [A,a,Y] | 165 ± 34 [A,a,X] | 122 ± 33 [A,a,X] | 60 ± 16 [A,a,Y] | 75 ± 24 [A,a,X] |
| | | +LT + M | 216 ± 37 [A,a,X] | 132 ± 23 [AB,a,X] | 106 ± 20 [B,a,X] | 128 ± 35 [A,a,X] | 58 ± 16 [AB,a,X] | 50 ± 16 [B,ab,X] |

A, B: Planting date difference. For each type of yield, least squares means with the same letter within a row indicates no statistical difference at α = 0.05. a, b: Insulation difference. For each type of yield, least squares means with the same letter within a column under the same pre-sprouting indicates no statistical difference at α = 0.05. X, Y: Pre-sprouting difference. For each type of yield, least squares means with the same letter within a column under the same insulation indicates no statistical difference at α = 0.05.

The average marketability (quality and speculation grades) of field stems was 49 ± 1%. Of the total yield across all treatments, 24 ± 1% of stems were quality grade, 25 ± 1% were speculation grade, and 51 ± 1% were cull grade. The proportions of quality and cull grade stems were affected by all main effects (*year, insulation, planting date, cultivar*, and *pre-sprouting*), while the proportion of speculation grade stems was only affected by the main effect of *year* ($p < 0.05$). In 2021, 33 ± 5% of stems were quality grade, compared to 14 ± 5% in 2022 ($p = 0.0375$). April plantings produced the least quality grade stems (18 ± 4%) compared to November (26 ± 4%) or March plantings (27 ± 4%; $p = 0.0032$). A total of 27 ± 3% of 'Galilee' stems were quality grade compared to 20 ± 3% of 'Carmel' stems ($p = 0.0001$), and 25 ± 3% of +PS stems were quality grade compared to 22 ± 3% of −PS stems ($p = 0.0225$).

### 3.5. High Tunnel Yield

Total and marketable yield were affected by the main effects of *pre-sprouting, planting date*, and *cultivar*, with marketable yield also affected by the main effect of *low tunnels* ($p < 0.05$). Significant interactions among these factors were also detected. The simple effects of *planting date, low tunnels*, and *pre-sprouting* at each level of the other factors for total and marketable yield are presented in Table 2. Average total high tunnel yields ranged from 117 to 369 stems per m$^2$ with an overall average yield of 254 ± 7 stems per m$^2$, while average marketable yields ranged from 32 to 280 stems per m$^2$ with an overall average of 142 ± 7 stems per m$^2$. Marketable yield was 50 ± 15 stems per m$^2$ greater in 2021 than in 2022 ($p = 0.0742$), 52 ± 13 stems per m$^2$ greater for +LT than for −LT ($p = 0.0231$), and 28 ± 12 stems per m$^2$ greater for +PS than for −PS ($p = 0.0207$). Marketable yield decreased

with each subsequent planting date, from $162 \pm 20$ stems per m$^2$ for November plantings to $128 \pm 16$ stems per m$^2$ for January plantings, $109 \pm 13$ stems per m$^2$ for February plantings, and $65 \pm 8$ stems per m$^2$ for March plantings. Marketable yield was $44 \pm 11$ stems per m$^2$ greater for 'Galilee' than for 'Carmel' ($p < 0.0001$).

**Table 2.** High tunnel anemone production by calculated mean ($\pm$SE) total and marketable yields in stems per m$^2$ by cultivar ('Carmel' and 'Galilee'), pre-sprouting (pre-sprouted +PS and non-pre-sprouted −PS), the presence (+LT) or absence (−LT) of low tunnels, and planting date (November, January, February, and March) in North Logan, UT, over 2020-21 and 2021-22.

| Cultivar | PS | LT | Total Yield (Stems per m$^2$) | | | | Marketable Yield (Stems per m$^2$) | | | |
|---|---|---|---|---|---|---|---|---|---|---|
| | | | **Planting Date** | | | | | | | |
| | | | November | January | February | March | November | January | February | March |
| 'Carmel' | +PS | −LT | 237 ± 30 [A,a,X] | 224 ± 28 [A,a,X] | 244 ± 31 [A,a,X] | 187 ± 24 [A,a,X] | 103 ± 27 [A,a,X] | 67 ± 18 [A,b,X] | 80 ± 21 [A,a,X] | 70 ± 18 [A,a,X] |
| | | +LT | 283 ± 36 [A,a,X] | 221 ± 28 [A,a,X] | 250 ± 32 [A,a,X] | 184 ± 23 [A,a,X] | 195 ± 51 [A,a,X] | 140 ± 36 [A,a,X] | 148 ± 39 [A,a,X] | 83 ± 22 [A,a,X] |
| | −PS | −LT | 226 ± 28 [A,a,X] | 213 ± 27 [A,a,X] | 172 ± 22 [A,a,Y] | 155 ± 20 [A,a,X] | 91 ± 24 [A,b,X] | 67 ± 18 [A,b,X] | 53 ± 14 [A,a,X] | 45 ± 12 [A,a,X] |
| | | +LT | 299 ± 38 [A,a,X] | 267 ± 34 [A,a,X] | 196 ± 25 [A,a,X] | 117 ± 15 [B,a,Y] | 204 ± 53 [A,a,X] | 163 ± 43 [A,a,X] | 104 ± 27 [A,a,X] | 32 ± 8 [B,a,Y] |
| 'Galilee' | +PS | −LT | 347 ± 44 [A,a,X] | 273 ± 34 [A,a,X] | 277 ± 35 [A,a,X] | 242 ± 31 [A,a,X] | 197 ± 51 [A,a,X] | 129 ± 34 [A,a,X] | 78 ± 21 [A,b,X] | 111 ± 29 [A,a,X] |
| | | +LT | 369 ± 46 [A,a,X] | 285 ± 36 [AB,a,X] | 320 ± 40 [A,a,X] | 187 ± 24 [B,a,X] | 280 ± 73 [A,a,X] | 200 ± 52 [AB,a,X] | 213 ± 56 [AB,a,X] | 95 ± 25 [B,a,X] |
| | −PS | −LT | 258 ± 32 [A,a,X] | 277 ± 35 [A,a,X] | 261 ± 33 [A,a,X] | 178 ± 22 [A,a,X] | 113 ± 30 [A,a,X] | 125 ± 33 [A,a,X] | 89 ± 23 [A,a,X] | 62 ± 16 [A,a,X] |
| | | +LT | 282 ± 36 [A,a,X] | 324 ± 41 [A,a,X] | 279 ± 35 [A,a,X] | 173 ± 22 [B,a,X] | 201 ± 53 [A,a,X] | 223 ± 58 [A,a,X] | 178 ± 47 [A,a,X] | 62 ± 16 [B,a,X] |

A, B: Planting date difference. For each type of yield, least squares means with the same letter within a row indicates no statistical difference at $\alpha = 0.05$. a, b: Low tunnel difference. For each type of yield, least squares means with the same letter within a column under the same pre-sprouting indicates no statistical difference at $\alpha = 0.05$. X, Y: Pre-sprouting difference. For each type of yield, least squares means with the same letter within a column under the same insulation indicates no statistical difference at $\alpha = 0.05$.

The average marketability of high tunnel stems was $52 \pm 2\%$. Of the total yield across all treatments, $26 \pm 1\%$ of stems were quality grade, $26 \pm 1\%$ were speculation grade, and $48 \pm 2\%$ were cull grade. The proportions of quality and cull grade stems were significantly affected by the main effects of *low tunnels, planting date*, and *cultivar*, as well as by the interaction effect of *low tunnels × planting date*. Additionally, the proportion of cull grade stems was significantly affected by the main effect of *year*, with $15 \pm 5\%$ more cull stems in 2022 than in 2021 ($p = 0.0496$). The proportion of speculation grade stems was not significantly affected by any main or interaction effects. The percentage of quality grade stems decreased with subsequent planting dates, from $35 \pm 3\%$ for November to $29 \pm 3\%$ for January, $23 \pm 3\%$ for February, and $19 \pm 3\%$ for March plantings. Of the total stems, quality grade stems made up $34 \pm 3\%$ for +LT compared to $18 \pm 3\%$ for −LT ($p = 0.0163$) and $29 \pm 2\%$ for 'Galilee' compared to $24 \pm 2\%$ for 'Carmel' ($p = 0.0009$).

*3.6. Marketing and Crop Value*

The estimated crop value across management practices ranged from $41 to $139 per m$^2$ in the high tunnel and $20 to $79 per m$^2$ in the field. High tunnel production costs consisted of a co-op delivery fee (42%), labor (34%), supplies (18%), and high tunnel construction (6%), compared to field production costs of labor (36%), a co-op delivery fee (36%), supplies (26%), and land (1%). Based on these costs, in the high tunnel, November plantings always produced positive returns, while January and February plantings only produced positive returns when low tunnels were used, and March plantings always produced negative returns. The greatest net returns of $38 per m$^2$ were calculated for a high tunnel November planting with pre-sprouting and low tunnels. Calculated field economic returns varied from negative $32 (April +LT + M + PS) to positive $5 per m$^2$ (November −LT + M + PS) across management practices, with only pre-sprouted November and March plantings with mulch resulting in positive economic returns.

*3.7. On-Farm Trials*

Of the 13 full sets of yield data submitted, the average yield (in stems per plant) was $6.7 \pm 2.1$ total and $4.7 \pm 1.5$ marketable for fall-planted tubers and $4.3 \pm 0.8$ total and $3.2 \pm 0.6$ marketable for spring-planted. Total yields ranged from 1.2 to 25.4 stems per plant and

marketable yields ranged from 0.9 to 20.3 stems per plant. Management practices varied by grower, with 23% of growers choosing to pre-sprout tubers. One grower used a high tunnel, while for fall plantings 15% of growers left tubers bare, 15% insulated with mulch alone, 23% insulated with a low tunnel alone, and 31% insulated with mulch and a low tunnel. Four growers noted extensive issues with rot, which occurred for fall and spring plantings with low tunnels and bare soil. The high tunnel grower (USDA hardiness zone 5b) did not pre-sprout and had marketable yields in the range of 4.2 to 20.3 stems per plant, while most of the field growers had marketable yields ranging from 1.0 to 6.5 stems per plant, with marketable yields generally increasing with increasing growing experience and fall-planted total yields generally increasing in warmer climates (hardiness zone). The average marketable yield was 1.6 stems per plant greater for fall plantings in 2021 than in 2022, and 0.6 stems per plant greater for spring plantings in 2022 than in 2021.

## 4. Discussion

In the high tunnel, the earliest harvest (2 March) began over two months before the average last frost date (15 May) and nearly five weeks earlier than the field (9 April). By allowing for vegetative growth throughout the winter and flowers beginning in early March, high tunnels maximized time at growing conditions in the optimal temperature range of 5 to 18 °C. As a result, the total high tunnel yield of 254 stems per m$^2$ (5.8 stems per plant) was in the upper range of the 3 to 7 stems per plant reported in other anemone production studies, whereas the total field yield of 151 stems per m$^2$ (3.4 stems per plant) was in the lower end of this range [6,8–10]. High tunnel production varied by year, with the average first harvest delayed by up to 12 days across planting dates and average marketable yield decreased by up to 69% in 2022 compared to 2021. Leaf and floral tissue were likely impacted by cold injury in 2022, as buds were first observed on 14 February and browning and necrosis were observed on some uncovered leaves after air temperature reached a minimum of −11.3 °C on the morning of 24 February.

Within the high tunnel, earlier plantings, low tunnels, and pre-sprouting further advanced harvest, giving plants up to 1.5 more months to flower before super-optimal temperatures (>25 °C) occurred, thus increasing marketable yield. Despite year-to-year variability, plantings from November through February showed similar marketable yield potential as the plants began flowering between March and May and finished flowering by early June. In contrast, the marketable yield of March plantings was 51% lower than all other planting dates as plants began flowering at the end of May, resulting in a window of less than a month before average daily temperatures reached 25 °C and plants began to senesce. Similar results were obtained in a Georgia field study (USDA hardiness zone 8) where total yield decreased significantly as planting dates progressed from November to February, with harvest for all planting dates ending the first week of May, when daily maximum air temperatures were in the range of 23 to 31 °C [10]. The harvest midpoint (T50) was staggered by one to two weeks for successive planting dates, indicating that growers could benefit from combining planting dates from November to February to produce a consistent supply of high-quality flowers for local markets.

In the high tunnel production system, first harvest was further advanced by up to two weeks with low tunnels and one week with pre-sprouting, which likely contributed to marketable yield increases of 61% with low tunnels and 29% with pre-sprouting. Similar findings were reported from a NY high tunnel trial (USDA hardiness zone 5), where anemone harvest was advanced by up to 16 days with low tunnels and 10 days with three weeks of pre-sprouting, resulting in yield increases of 23% with low tunnels and 28% with pre-sprouting [8]. Likewise, the earliest UT harvest of 2 March occurred from fall planting on 16 November (106 days to harvest), while the earliest NY harvest of 14 April occurred from winter planting on 23 December (112 days to harvest) [7]. This shows the non-growing season provided up to 3.5 months for early vegetative growth, and fall planting further advanced flowering compared to winter plantings. Low tunnels advanced harvest more than pre-sprouting for November and January plantings, while

pre-sprouting advanced harvest more than low tunnels for February and March plantings, indicating that the flowering of earlier plantings is likely limited by sub-optimal (<5 °C) temperatures, as found in ranunculus [50]. Low tunnels also improved the proportion of quality stems by 86% compared to plants with no additional cover, which contributed to the greater marketable yields observed, and could be the result of the 2 to 3 °C daily average air temperature lift provided by the low tunnels and/or reduced irradiance under the low tunnels.

Earlier field plantings also had more time to flower before temperatures became super-optimal, which advanced first harvest and harvest midpoint dates and increased yields compared to later plantings. For November and March plantings, insulation of any type improved average marketable yield from 35 to 86 stems per $m^2$ by reducing cold injury. Meanwhile, April plantings, with an average marketable yield of 44 stems $m^2$, began flowering in early June, when maximum daily air temperatures were already approaching 25 °C, and were likely limited by heat. Emergence for uninsulated November plantings varied from 69% in 2020-21 to 41% in 2021-22, likely from the timing of the minimum soil temperatures. In 2020-21, the bare soil temperature fell below freezing within five days of planting and reached a minimum of −9.2 °C on 9 December, likely before root growth occurred. In contrast, the bare soil temperature stayed above freezing until early January in 2021-22, allowing two months of potential root growth before soil temperature fell below freezing from mid-January to early March and reached a minimum of −6.9 °C on 24 February that likely resulted in cold injury. Though the yield of fall plantings was improved by all insulation types, soil temperature fluctuated the most and reached a minimum of −3.8 °C with low tunnels alone, indicating that mulch buffered soil heat losses and may reduce risk from year-to-year weather variability [23].

Both high tunnel and field production were improved with the use of the cultivar 'Galilee,' which increased marketable yield, on average, by 32% in the field and 47% in the high tunnel compared to 'Carmel.' Advertised stem lengths are 46 to 91 cm for 'Galilee' and 41 cm for 'Carmel' [51]. The longest stem in our study was 46 cm from 'Galilee', which produced significantly more quality grade (>25 cm length) stems than 'Carmel' yet still only produced less than 30% of stems that were quality grade, a grade with a length threshold roughly half of advertised lengths. Moreover, 'Galilee' is marketed as producing many blooms that are smaller in diameter while 'Carmel' is marketed as producing fewer blooms that are larger in diameter [51]. While we did not measure bloom diameter, we did not observe any visual differences in bloom size and sold both cultivars at the same price, indicating that 'Galilee' is a better choice for production due to its higher marketable yields and longer stem lengths.

Anemone production was limited by short stem lengths, and hence, low marketability, as also noted within our on-farm trials. Approximately 48% of high tunnel and 51% of field stems were graded as culls because of their short (<20 cm) stem length. Similarly, anemone stem lengths in other US research trials ranged from 10 to 30 cm [8–10], while in an Israeli high tunnel study using 35% shade, approximately 30 to 40% of total stems were longer than 40 cm [6]. Stem length is influenced by a variety of factors, such as cultivar, irradiance, temperature, wind, and flower timing [9,10,14,36]. In this study, 30% shade cloth was selected to increase stem length without jeopardizing yield, but increasing the shade percentage may further increase anemone stem lengths. In a Georgia field study (USDA hardiness zone 8), shading anemone 'De Caen' plants with 67% shade cloth increased stem length by 5 to 10 cm with no change in total yield, but stem lengths only reached a maximum of 30 cm and stem diameter was not reported [9]. Additionally, soaking anemone tubers in gibberellic acid before planting increased stem lengths by 5 to 10 cm in a greenhouse study and warrants further investigation in a field setting [52].

Net economic returns were largely dependent on quality grade yield since the calculated costs of production were similar across management practices and speculation grade stems were less likely to sell. November planting with pre-sprouting maximized quality grade yield and resulted in net returns up to $38 per $m^2$, which is in the upper

middle range of other high tunnel cut flowers, such as peony, snapdragon, and ranunculus that yielded net returns of $18, $28, and $54 per m$^2$, respectively [12,53,54]. Moreover, the earliest anemone flower on 2 March was the earliest flowering of a cool-season cut flower recorded in North Logan, UT, high tunnel trials (Melanie Stock, unpublished data), providing growers an opportunity to make use of a nontraditional production window and plant a second crop in the same tunnel [35]. For example, planting a warm season crop (e.g., celosia) in June would maximize the use of limited space and generate additional returns, further increasing profitability for small farms. While field anemone production improved using insulated fall or early spring plantings, only November and March plantings with pre-sprouting yielded enough quality and speculation grade stems to generate positive economic returns with local wholesale markets. Additionally, anemones were a less popular purchase among florists than other cool season cut flowers, such as tulips, ranunculus, and peonies, which could be a short-term trend or indicative of lower market demand. However, social media accounts of local farms indicate that anemones are a popular choice among growers who use retail outlets like farmer's markets, CSAs, social media, and on-farm events to market shorter stems with higher markups. Prioritizing retail markets is recommended to maximize anemone profitability and utilize high total yields regardless of wholesale marketability.

## 5. Conclusions

Planting pre-sprouted anemone tubers with low tunnels in November delivered the earliest harvests and greatest yields, stem quality, and net returns in a USDA hardiness zone 5 high tunnel production system. Succession planting of 'Galilee' in high tunnels from November through February may allow growers to produce a consistent supply of flowers from early March to the end of May before planting a warm season crop in June. High tunnel plantings after mid-February are not recommended as harvests beginning in mid-May or later are limited by super-optimal air temperatures by late June. For growers lacking the space for a high tunnel, combining pre-sprouted, mulch-insulated November and March plantings supplies smaller yields of anemone from the beginning of May to the end of June. Anemone production in the US Intermountain West is mainly limited by short stem lengths, making it an ideal crop for retail sales outlets, and additional research exploring how to increase anemone stem lengths in high tunnel and field production systems is recommended.

**Author Contributions:** Conceptualization, S.R. and M.S.; Data curation, S.R.; Formal analysis, S.R., M.S., X.D. and R.W.; Funding acquisition, M.S., B.B. and D.D.; Investigation, S.R. and M.S.; Methodology, S.R., M.S. and X.D.; Project administration, M.S.; Resources, M.S., B.B. and D.D.; Supervision, M.S.; Visualization, S.R., M.S. and X.D.; Writing—original draft, S.R.; Writing—review & editing, M.S., B.B., D.D., X.D. and R.W. All authors have read and agreed to the published version of the manuscript.

**Funding:** This research was funded by a USDA-NIFA Specialty Crop Block Grant administered by the Utah Department of Agriculture, the Association of Specialty Cut Flower Growers, a United States Department of Agriculture (USDA) National Needs Fellowship, and by the Utah Agricultural Experiment Station, Utah State University, journal paper number UAES #9630.

**Data Availability Statement:** Data are contained within the article.

**Acknowledgments:** We thank James Frisby of Utah State University and our undergraduate students, Kasey Battson, Anna Collins, and Olive Stewart, for their technical assistance.

**Conflicts of Interest:** The authors declare no conflict of interest. The funders had no role in the design of the study; in the collection, analyses, or interpretation of data; in the writing of the manuscript; or in the decision to publish the results.

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
