# Peer review of "Anemone Cut Flower Timing, Yield, and Quality in a High-Elevation Field and High Tunnel"

_horticulturae, doi:10.3390/horticulturae9010002_

Round 1

Reviewer 1 Report

Overall, the paper is good. The only comment that I have is that the Acclima TDR or similar Campbell Scientific TDR probes are not accurate at a depth of 0.05 m.

Reviewer 2 Report

This manuscript is well written. However, there are also the following problems:
1. The cultivation status of the soil (such as tilling depth, crushed soil, humidity and fertility parameters annual changes) was not clearly explained. We know it is one of the most important factors affecting crop growth, and perhaps the differences in plant growth mentioned in the manuscript are mainly due to soil status.
2. The   test data should be further analyzed. For example, the relationship among cumulative marketable yield and main soil temperature and air temperature is best to have a regression
data analysis.

Reviewer 3 Report

The study considered the technological evaluation of Anemone Cut Flower Timing, Yield and Quality in a High-Elevation Field and High Tunnel. The manuscript is well done; in many situations the research has a local impact.

1.       Grammatically corrected

2.       The abstract should be redone because the aim is not well defined. The objectives of the research are not the same as the aim for which it is done and which should generally solve a problem.

3.       In general, abbreviated terms were not used in the abstract

4.       In discussions, 2-3 working hypotheses; development opportunities must be highlighted

5.       It would have been very good if, from a statistical point of view, a correlation could have been made between the climatic conditions, the crop year and the yield.

6.       The bibliographic references are very old, not accepted for 2023 paper

7.       The bibliography must be rearranged according to the journal or unpublished bibliography, e.g. Stock, M. Unpublished Data. 2022
